# Analysis of Crack Width Development in Reinforced Concrete Beams

**DOI:** 10.3390/ma14113043

**Published:** 2021-06-03

**Authors:** Barbara Goszczyńska, Wiesław Trąmpczyński, Justyna Tworzewska

**Affiliations:** Faculty of Civil Engineering and Architecture, Kielce University of Technology, Al. Tysiąclecia Państwa Polskiego 7, 25-314 Kielce, Poland; bgoszczynska@tu.kielce.pl (B.G.); j.tworzewska@tu.kielce.pl (J.T.)

**Keywords:** reinforced concrete, crack initiation and development, monotonic and cyclic load, simple description of crack width, crack width calculation, digital image correlation

## Abstract

The reliability and durability of reinforced concrete structures depend on the amount of concrete cracking. The risk associated with cracks generates a need for diagnostic methods for the evaluation of reinforced concrete structures. This paper presents the results of a study of 10 single-span reinforced concrete beams to follow the process of crack formation and changes in their width. The beams were loaded to failure with two forces in a monotonic manner with unloading and in a cyclic manner. Continuous observation of the crack formation process was provided by the digital image correlation system. The simplified method for estimating the maximum crack width is proposed. The presented results confirmed the stochastic character of the process of crack formation and development. The maximum crack widths calculated on the basis of the proposed formula were on the safe side in relation to those calculated according to Eurocode 2. It was also confirmed that the distances between cracks do not depend on the loading manner. Hence the density function describing the distribution of distances between cracks can be used to assess the condition of reinforced concrete elements. The research has also shown the suitability of the DIC system (ARAMIS) for testing concrete elements.

## 1. Introduction

Concrete is the most extensively used material in the construction process, with the annual global consumption level at 4.2 billion tonnes by volume [1]. Its properties are constantly enhanced, allowing the erection of very-complex buildings and engineering structures.

The main benefits of this composite material include strength, durability, shock resistance, relatively low deformability, high fire resistance, and resistance to temperature and moisture fluctuations. It provides a high degree of stiffness and is easy to make into any shape. These properties make concrete a safe and durable option for various construction purposes—from civil engineering to residential to hydraulic engineering projects.

Reinforced concrete (RC) structures, most common among the structures built from concrete, are exposed to static and dynamic loads, extreme weather and cyclic temperature changes. As a result of physical, chemical, mechanical, physicochemical and biological impacts, the structural materials used in making RC structures deteriorate, thereby shortening their safe use.

Advances in technology have resulted in a trend towards the use of increasingly higher strength concrete and steel. Economical use of high quality and high ductility steel leads to correspondingly larger deformations, contributing to an increase in displacements and crack widths. These effects are not relevant for the load-bearing capacity of reinforced concrete elements, but they have a decisive influence on the structure’s performance, durability included. Not surprisingly, with the changing performance characteristics of concrete and steel, the state of cracking is still a subject of research.

The multitude of factors contribute to concrete cracking [2,3,4,5,6,7,8,9]. The heterogeneity of materials and the complex state of stresses and strains accompanying the crack formation make the descriptions of these phenomena be constantly modified, especially from a practical point of view. There are two approaches for the study of concrete cracking.

One relies on the fracture mechanics and concrete resistance modeling of concrete [10,11,12,13,14,15,16,17] which considers the concrete mix components, internal damage, stresses and temperature.

The second approach, applied in this paper, involves tracking the changes in crack formation and crack width to develop diagnostic methods for the evaluation of reinforced concrete structures whose condition depends on the intensity of cracking. Cracking intensity is the number of cracks per unit length a member and the crack width which determines the reinforcement corrosion risk. Random variables influence the crack width size; hence the crack width is also a random variable of which the probabilistic or the evaluation properties follow a normal or log-normal distribution. In terms of the limit states, In the limit states view, the maximum crack width is a quantile at β, where the probability of this event β is fixed a priori and can be a function of the purpose and durability of the structure. From a practical point of view, of particular importance is the maximum crack width estimation accuracy, which decides the risk of corrosion [18]. Corrosion significantly affects the durability of RC structures as it is the leading cause of their deterioration [6,19,20].

Although various crack width estimation methods have been developed over the years, research needs to be continued [4,5,6,7,13,21]. The proposed relationships are fairly complicated and largely based on many empirical coefficients that take account of concrete composition and strength characteristics, way of curing, loading scheme, cover thickness, type and diameter of reinforcing bars. New and simplified techniques for crack width estimation must be found, particularly reflecting the stochastic nature of the crack formation process [2]. The search for the new methods is aided by modern apparatus [3,4,22,23] that allows continuous monitoring of crack propagation and measuring crack width under dynamic loads.

This paper presents the results of a broad-range program of testing reinforced concrete beams subjected to monotonic loading with unloading and cyclic loading. The research was conducted with a view to developing a method of diagnosing reinforced concrete bridges.

In parallel, work was carried out on diagnostics using acoustic emission technique [24,25].

Since the durability of reinforced concrete structures depends on the intensity of cracking and crack widths, a parallel topic of tracking the process of crack formation and development was undertaken. A known density function for inter-crack distances and crack widths under loading will allow the development of a method for RC structure condition diagnosis based on artificial intelligence. A diagnosis that relies on methods, i.e., acoustic emission and artificial intelligence will be much more reliable, particularly in difficult cases.

The formation of cracks and their development was recorded using the ARAMIS digital image correlation (DIC) system. The system provides a 3D real time display of strain field overlaid on live image, thereby allowing a continuous measurement of crack widths. The use of a power supply system with a controller, which was synchronized with the measuring equipment, enabled the implementation of cyclic loads.

The crack width test results were compared with those calculated according to EC2 [26] and with the results of the estimated crack width perpendicular to the element axis obtained with the simplified formula. The simplified formula is based on the previously proposed probabilistic model of the inter-crack fields [27]. It was demonstrated that the crack formation is a stochastic process and that the maximum width can be described by a relatively simple, loading mode independent relationship.

## 2. Test Program

### 2.1. Beams and Beam Designation

Ten reinforced concrete (RC) beams manufactured at the prefabrication plant producing bridge elements were subjected to bending tests. The beams were simply supported and had two concentrated loads applied at a distance of 1 m from the supports. The beams, each 120 mm × 300 mm in cross section and 3300 mm in length (3000 mm effective span), were ordered from concrete C40/50, and the bars were made of B500S (class B) steel. The longitudinal reinforcement ratio at the beam tension zone was 2%. Four of the beams were designed without stirrups, and two had no compression reinforcement in the central section. Different loading regimes were applied: monotonically increasing load until failure, loading and unloading and low-cycle loading. The tested members are shown in Table 1. Beam designation followed the pattern shown in Figure 1.

Material composition and properties of the concrete mixture obtained from prefabrication plant for concrete grade C40/50:Cement CEM I 52,5N HSR NA Chełm—390 kgWater-cement ratio 0.4Water—155 kgAggregates—basalt 8/16–694 kg (35.2%), basalt 2/8–617 kg (31.3%), river sand 0/2–660 kg (33.5%)Superplasticizer—1.84 kgAir entraining agent—0.47 kg.

The beam reinforcement design with or without longitudinal bars at the compression zone and with or without stirrups in the midspan section of the beam is illustrated in Figure 2, Figure 3 and Figure 4.

### 2.2. Loading Diagram and Program

Figure 5 shows a static scheme of beams A2, C2 and D2, for which the reinforcement and load applied were selected so as to induce failure due to bending moment.

Three different loading regimes were used, marked as M—monotonic loading progressively increased to failure, O—loading and unloading, and C—low-cycle loading, as compiled with loading rates in Table 2. The monotonic loading of beams A2, C2 and D2 (two beams of each type) is illustrated in Figure 6. The loading time indicated in Figure 6 corresponds to the tests on beams A2M. Six phases of loading/unloading were used in this study, as shown in Figure 7. The first five phases included ten unloading cycles in each phase at five different loading levels, whereas in phase 6, the load was increasing to failure. Figure 8 illustrates the low-cycle loading performed in four phases. The first three phases included 100,000 cycles per phase at three loading forces: 10, 40 and 58 kN. In phase 4, the beam was monotonically loaded to failure. The loading program in particular cycle phases (Figure 8) was implemented by alternatively increasing the loads from 5 kN up to the maximum value at a given phase.

### 2.3. Additional Tests—Strength of Concrete

The strength of the concrete was tested on the 150 mm × 150 mm × 150 mm cuboids fabricated while casting particular concrete beam pairs in the multi-cavity mould. The failure force test results for the concrete cubes, the calculated compressive strength of the concrete and the relevant statistics are presented in Table 3 and Table 4, respectively.

Table 5 compiles the results of calculations for individual pairs of beams:Average concrete compressive strength estimated according to (1)
(1)$$f_{\mathrm{cm}}^{\mathrm{CYL}}=0.8f_{\mathrm{cm}}^{\mathrm{cube}}$$Characteristic concrete compressive strength according to (2)
(2)$$f_{\mathrm{ck}}=f_{\mathrm{cm}}^{\mathrm{CYL}}-8$$Class of concreteModulus of elasticity according to (3)
(3)$$E_{\mathrm{cm}}=22{\left(0.1f_{\mathrm{cm}}^{\mathrm{CYL}}\right)}^{0.3}$$

For example, Figure 9 shows the force–strain curves automatically obtained while testing the concrete specimens corresponding to the A2M and A2O beams in the Zwick hydraulic press 6000. For each casting, six cubes were made and then destroyed. The numbers 1,2,3,4,5 and 6 are the specimen numbers. As can be seen from the graph, the ultimate strain amounts to approximately 0.85% on average, which means much greater ultimate strains than those adopted for the concrete class below C50/60, amounting to approximately 0.35% [26]. The force–strain relationships obtained during the tests on the remaining beams are very similar to those shown in Figure 9.

### 2.4. Additional Tests—Strength of Steel Bars

A total of 124 bars used for the main reinforcement were examined, including 58 bars of ϕ 12 mm and 66 of ϕ 14 mm. Table 6 summarizes the tensile test results and the statistics for the results of the reinforcing bars. The table also shows the automatically obtained results of the elastic modulus E, yield strength R_p0.2_, upper yield point R_eH_ and tensile strength R_m_.

An example of the stress–strain relationships obtained during the tensile test of BS500 steel bars is shown in Figure 10 and Figure 11. It follows from the graphs that the reinforcing bars were made of high ductility steel. This high ductility of steel distinguishes the currently used steel grades from those used previously, whose brittleness increased with an increase in strength, and even the elastic limit was assumed.

## 3. Test Setup and Instrumentation

### 3.1. Test Setup

The test setup for the reinforced concrete beams was as shown in Figure 12. The beams were supported by bridge bearings, of which one was fixed, and the loads were imposed automatically by two independent hydraulic actuators of 0–400 kN loading capacity.

During the loading, displacements were continuously measured using the HBM measuring system and the field deformation of the middle section of the beam (between forces) by the Digital Image Correlation (DIC) system. The HBM—Hottinger Baldwin Messtechnik system was used to record deflections in the midspan of the beam, under forces and near the supports, while the DIC system was used to measure strains in three directions of the previously prepared central surface, including cracking on the side surface of the beams. The actuator’s analogue outputs allowed synchronizing the entire test apparatus, and the recorded force was used to control the HBM and DIC systems measurements. Apparatus synchronization enabled assigning deflection results and the image of strains within the tested field for each force value in the middle section of the beam under load.

### 3.2. Crack Width Measurements with DIC System

DIC is widely used in many fields of science, e.g., biomechanics, astronautics, microelectronics and now also in civil engineering [28,29,30,31,32,33,34,35,36,37,38]. In the experiments, the commercial ARAMIS system was used which is presented in detail in [19,39]. During loading, strains were recorded using the DIC system in the area, as shown in Figure 13. The area between forces was analyzed. The test surface had to be prepared by applying a black paint pattern to enable strain tracking.

Analysis of the cracking process in tests conducted with the DIC system is performed after the test has been performed, using the results of calculations for subsequent images presented in the form of strain maps. The system is used for non-contact three-dimensional strain measurements. It performs calculations and documents deformations. The system uses a series of images from two digital cameras on one tripod. It recognizes the surface texture of the sample area in the images. The first photo in the series is treated as a photo of the object before loading and is used as a reference “object”. After taking the measurements, the system generates a grid on the first photo. The grid consists of small square or rectangular planes called facets. It assigns each facet a single and unique texture of grey dots. The unique texture makes it possible to find facets in subsequent images. Based on the similarity of the facets in the successive images, the strain of the facets is calculated.

Consecutive images in the form of strain distribution maps are used in the analysis of the cracking process. Figure 14 and Figure 15 include strain map examples for the middle section of the 1000 mm long beam C2M-1 at the load level of 0.75 and 0.95, respectively. The strains on the maps are marked using a colour scale (the log symbol means ln and expresses the fractal measure of deformation).

Analysis of the strain map indicates local strain accumulations at the lower zone of the beam in tension, identified as cracks in concrete (brittle material).

The measurement of the crack width (along the reinforcement centroid line) included computing the change in the length of section ΔL at a random force level [19,39] throughout the loading process. The measurement scheme is shown in Figure 16. The crack has to be defined in that measurement method to distinguish cracks from surface microcracks and optical noises. In paper [19], a crack was defined as the section length increment of at least ΔL_i_ ≥ 0.05 mm.

To illustrate the correctness of the calculated crack widths, Figure 17 compares the crack width measurement results obtained from the traditional method (with a Brinell magnifying glass) and the DIC system. The test was performed on an additional test beam loaded monotonically to failure with breaks made for measurements with the Brinell magnifier—Figure 18.

The results obtained with the magnifier and the DIC system fall within the magnifier’s error area. Measurement uncertainty, described in [40], is ±0.028 mm. In order to confirm the results, the results of deflections obtained from the HBM measuring system and the DIC system were compared and found to be virtually the same [41].

This satisfactory agreement made it possible to use the DIC system to record and measure the quantities describing the cracking state of the beams based on consecutive images. The strain measurements were recorded over the entire prepared area of the beam, thereby enabling the measurement of distances between the cracks and crack widths at the height corresponding to the centroid of the tension reinforcement during loading to failure. The frequency of image triggering events was adjusted to each loading program. The images were captured either at a specified time or at appropriate load values. A single tripod with two cameras was used to capture images every 30 s for A2M beams and every 20 s for C2M and D2M beams.

When testing A2O beams, the so-called “triggerlist” was used, and a program for taking images triggered by a defined voltage corresponding to a given load was written. The photos were taken with one DIC system tripod with two cameras at defined force levels, as shown in Figure 19, where the dots represent the images taken. During phase 6, i.e., with the load gradually increasing to failure, the images were captured every 10 s.

For beams A2C, the images were captured intermittently due to the limited system memory. In this way, with each image, one actuator achieved the maximum load for a given phase, and the other actuator—the minimum load, because the actuators worked with a phase shift, i.e., alternately, as shown in Figure 20. Then there was a break of 1500s, after which the process repeated. During phase 4 of monotonic loading to failure, images were taken at intervals of 30 s.

## 4. Test Results

Three quantities typically characterize the cracking state. These are the cracking moment M_cr_, the distance between the cracks s_r_ and the crack width w. The last two of these quantities are functions of primarily the bending moment M, which can be written as:s_r_ = f (M)(4)
(5)$$w=f\left(M\right)$$

All these characteristics are interrelated. It can easily be shown that the cracking moment is a special property of these functions (4) and (5), because for M < M_cr_, the distance between the cracks would be greater than the beam length, and the crack width would be zero. The crack width and the distance between the cracks are thus strongly interrelated. Accordingly, the results of the RC beam test are presented for these three quantities.

The use of the DIC system in this study enabled the analysis of crack formation and development processes (cracking moment, crack width and distance between the cracks) for the loading program specified in Table 2.

Figure 21 and Figure 22 show an image the cracking state on the A2M-2 beam at the load level of 0.75 and 0.95, respectively, failure load. The calculation mask (deformation map) applied to the side surface of the beam after tests allows the observation of cracks at any selected load level, as illustrated in Figure 21 and Figure 22 (crack numbers are added in the middle section of the 1000 mm long beam).

As an example, Table 7 summarizes the crack width measurement results for cracks number 4, 5 … 13 (at the height corresponding to the centroid of tension reinforcement) for the A2M-2 beam, calculated from the consecutive images numbered 212 to 224. The sign X indicates the location of the crack on the beam’s middle section between the loads, where the moment is constant or approximately constant. Table 8 shows the results of the crack width measurements at the load level of 0.75 and 0.95 of the actual damage force.

The history of crack formation versus the bending moment in the middle segment X = 1000 mm in length on the A2M-2 beam side surface is shown in Figure 23 and Figure 24. The vertical lines in Figure 23 illustrate the cracks formed at various loading levels (bending moments). The distances between the vertical lines correspond to those between the cracks at a given load. The number of cracks equals the number of vertical lines. Crack density refers to the number of cracks per 1 m.

Deterministic crack formation criteria lead to a description in which all cracks are formed simultaneously over the entire section of the beam, where the moment reaches the critical value for cracking. The cracking on the middle section of the beam, between the forces, proceeds gradually, despite the constant value of the bending moment in this section, and new cracks appear even at the load that significantly exceeds the cracking load. Therefore, it can be concluded that cracks in the beam subjected to pure bending action under increasing load initiate successively as a result of a random spread of strength and forming limit in tension, which proves that both the distance between cracks and crack width are, in fact, stochastic processes. The deterministic description of the distance between the cracks can only refer to the final state of the cracking process.

The relationship in Figure 23 was used to calculate the distances between consecutive cracks, listed (for the A2M-2 beam) in Figure 24. The calculated mean inter-crack distance was taken.

A crack formation diagram similar to that in Figure 23 is shown in Figure 25 for the A2M-1 beam, and in Figure 26 and Figure 27 for cyclically loaded A2C-1 and A2C-2 beams, respectively.

The results confirm that the crack formation and development are stochastic, regardless of the loading program. The non-parametric tests (U Mann–Whitney or Kruskal–Wallis) used to analyze the results obtained, lead to the conclusion that there is no correlation between crack distances and loading method.

The crack formation graphs were used to determine the third quantity describing the cracking state, i.e., the cracking moment. Table 9 compiles the values and location of the first crack in eight beams tested, and the calculated relative cracking moment value against the breaking/failure moment. The scatter of the results is relatively high.

Analysis of the results confirms that the crack width and the distance between the cracks depend on the moment value, while the cracking moment is not moment-dependent. A small cracking moment in beams A2M-1, A2M-2 and A2O-1 proves that during concrete hardening, some action (e.g., concrete demoulded too early) resulted in the formation of an internal network of micro-cracks in its structure. Therefore, immediately after the load was applied, the internal microcracks developed into cracks visible on the side surface of the beams. This shows that many factors, some difficult to establish, affect the formation of cracks in concrete elements and that it is essential to test early age concrete.

## 5. Crack Width Calculation

Crack width is a crucial characteristic for the cracking limit state because it is the crack width, or rather its maximum value that decides the durability of concrete structures. Checking the cracking limit state of a structure consists of calculating whether cracks will appear in the structure under the influence of loads or, more generally, actions on the structure, or whether the crack width exceeds the permissible size. Thereby the calculations involve the estimation of the maximum value, which, as it depends on many factors, is characterized by a substantial random scatter and thus, the estimate will be more or less accurate. From a practical point of view, it is therefore important to estimate the maximum crack width in a safe and relatively simple manner.

As proposed in [27], the probabilistic under-load behavior model for the area between the cracks, shown in Figure 28, enables the formulation of a simplified formula for calculating the crack widths due to bending perpendicular to the axis of the element. In this model [27], in order to take into account the influence of existing cracks on the formation of new ones, it was assumed that in parts left and right of the crack (a′ and a″, Figure 28), the decrease in concrete stress excludes the formation of a new crack. It follows that the strain difference between steel and concrete can only occur in this section. It is called the passive section of length 2aω. Thus, the crack width can be calculated from Formula (6).
(6)$$w=2{a\omega \varepsilon }_{s}$$
where:
w—crack width,a′, a″—passive section measured from the cross-section with the crack,2a = a′ + a″ε_s_—steel strain in the section with the crack,ω ≤ 1—geometric factor for steel and concrete strain differences in section **a**.

The size of the area surrounding the crack, where stress decreases, is generally a random function. However, in search of a simplified formula, the continuous changes of the crack width were measured using the DIC system. The measurement of the crack width development under loading indicates the linear increase of up to 0.9 of the failure force [42], as shown by examples of cracks no. 2 and 14 (Figure 29) on beam D2M-2 and crack no. 11 on beam C2M-2 (Figure 30). This was confirmed by analyzing the average crack width increase for individual beams, as shown in Figure 31. It can thus be concluded that the course of changes in the section 2aω length can be assumed as linear.

Known crack width enables the estimation of the 2aω value which is the product of the length of the bond loss section 2a = a′ + a″ and the geometric factor for strain differences between steel and concrete ω≤1.
(7)$$2a\omega =\frac{w}{\varepsilon_{s}}$$
where:$$\varepsilon_{s}=\frac{M}{{\mathrm{zA}}_{s1}E_{s}}$$
in which:
M—bending moment valueA_s1_—cross sectional area of tension reinforcementE_s_—steel modulus of elasticityz—lever arm of internal forces of the flexure section.

The lever arm of internal forces in cracked flexure sections does not change significantly in the loading process, so, without much error, it is possible to take this value as a constant and calculate it based on the work of the section in bending in phase IIa (cracked cross section, linear plot of stresses in compressed concrete). Since BS500 steel reinforcement was used in the compression zone of the tested RC beams, the maximum internal force arm was assumed to be:$$z=d-d_{2}$$
where:
d—effective depth of a cross-sectiond_2_—distance of compression edge of the beam section to the centroid of the compression reinforcement.

The maximum crack width was obtained from the DIC system at selected load levels for the tested five beams with a constant moment in the central segment, and the value of **2aω** was calculated from Formula (7). The results of the calculations are given in Table 10 together with the fixed maximum, minimum and mean values.

The obtained results are presented graphically in Figure 32. The points in the figure correspond to the calculated mean, maximum and minimum value of 2aω, and the solid line is described by Relationship (8) obtained using the least squares method (LSM).
(8)$$2a\omega =-30.53\frac{M}{M_{\max}}+110.50$$

When analyzing the Figure 32, it can be noticed that the point with the coordinates [0.83; 127.04] is the farthest from the solid line obtained by LSM. Considering that the durability of the concrete structure is decided by the cracks with the maximum width, crack width estimations must give safe results. Hence, the solid line obtained by LSM was shifted by the vector υ [0; 41.88] and intersects point with coordinates [0.83; 127.04], as shown in Figure 33. To stay on the safe side, to determine the bond loss segment section, relationship (9) was adopted.
(9)$$2a\omega =-30.53\frac{M}{M_{\max}}+152.38$$

Table 11 compiles the results of crack width calculations from Formula (6) at selected load levels with the use of Relationship (9).

Figure 34 compares the experimental results, the results calculated according to the simplified Formula (6), assuming Relationship (9), and the results of the maximum crack width calculations to EC2. The horizontal line in the drawing corresponds to the permissible crack width of 0.3 mm.

Analysis shows that estimating the crack width from the simplified Formula (6), assuming the value of 2aω from Dependence (9) based on the concept of the probabilistic model of crack formation presented in Figure 28, allows a safe-sided estimation of the maximum crack width. This estimation method is more consistent with the experimental results and is on the safe side compared to the estimation based on the formulas given in EC2.

## 6. Discussion

The reliability and durability of a reinforced concrete element is closely related to the extent and severity of concrete cracking, which has a profound effect on reinforcement protection against corrosion. Concrete cracking research follows two approaches.

One approach is based on the fracture mechanics and modeling of concrete resistance including cracking. This approach looks at the influence of concrete mix components, aggregate size, internal mage, additives used and stress. Fracture mechanics and fracture toughness modeling have been covered in many papers [10,11,12,13,14,15,16,17]. Development of these issues is very important for numerical modeling of the performance of concrete elements. Their practical application is in design, including proper design of reinforcement.

The second approach, which is presented in this paper, is the tracking of crack pattern and crack width changes, utilized toward the development of methods for the assessment of reinforced concrete structures. The process of the loading of reinforced concrete beams is accompanied by the process of crack formation. As the load increases, the initially rapid changes in the crack pattern decrease and the pattern gradually stabilizes. The crack formation process depends on many factors, but for a given element it is a function of the load. It is thus a function of one variable, and the quantities such as shape, section dimensions, type of reinforcing steel, concrete mix composition, etc., act as parameters. The combined effect of these factors determines the intensity of the process which is a function of load. The crack intensity is the number of cracks per unit length of the element and the crack width, which determines the corrosion protection of the reinforcing bars. The paper presents the results of an extensive research program on reinforced concrete beams subjected to monotonic loads, and cyclic loads. The research presented in this paper was carried out with a view to developing a method for diagnosing reinforced concrete bridges; in particular, using the acoustic emission method [24,25]. Since the durability of reinforced concrete structures depends on the intensity of cracking and crack width, a parallel study was undertaken concerning the tracing of the crack formation process and development. Knowledge of the density function describing the distribution of distances between cracks in the process of loading and changes in the crack width would provide a reliable method of assessing the RC structure condition based on artificial intelligence. Topics related to the determination of the density function describing the distribution of distances between cracks have been addressed previously [2]. However, hardware possibilities in the field of computer capacity as well as testing equipment in the field of alternating loads and continuous crack width observation available in those time have stopped further development. The use of a program able element loading controller and an ARAMIS optical system for examining the lateral surface (1.2 m × 0.30 m) of the beam between the forces, enabled the continuation of the earlier subject matter. The results of the study of 10 reinforced concrete beams loaded according to 3 programs of monotonic loading to failure, with unloads loads, and cyclic loading gave a quantitative sufficient set of data to draw conclusions from the analyses. It was found that:The obtained results of continuous measurement of crack width changes during the loading process allowed the formulation of a simplified formula for estimating the maximum crack width;The maximum crack widths estimated on the basis of the proposed simple relation were on the safe side in relation to those calculated according to the Eurocode 2;The results of the analysis of the distances between cracks developed in the form of drawings and tables illustrate that the formation and development of cracks is a stochastic process, and also allows the conclusion that there is no relationship between the distances between the cracks and the method of loading;The optical system consisted of a tripod with an arm (distance between cameras) that ensured the observation of surfaces of dimensions 1.5 m × 1.0 m for use in the study of the process of crack formation and development in reinforced concrete beams.

The obtained results are the basis for further work, that will include consideration of diagonal cracks that arise at the section of the combined action of the bending moment and transverse forces, and the preparation of the experimental function of crack density and its verification.

## Figures and Tables

**Figure 1 materials-14-03043-f001:**
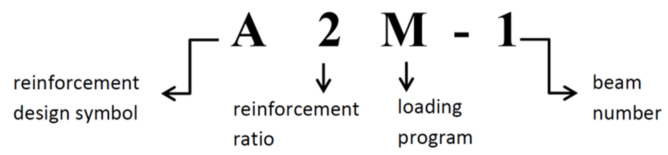
Beam designation key.

**Figure 2 materials-14-03043-f002:**
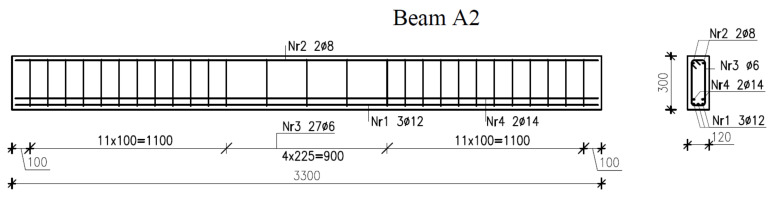
Reinforcement in beams A2.

**Figure 3 materials-14-03043-f003:**
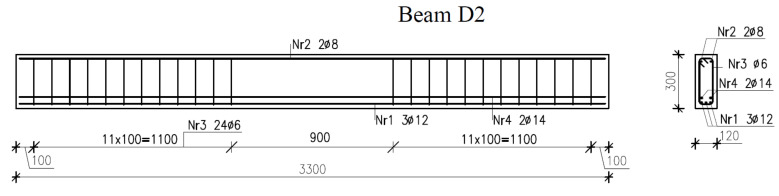
Reinforcement in beams D2.

**Figure 4 materials-14-03043-f004:**
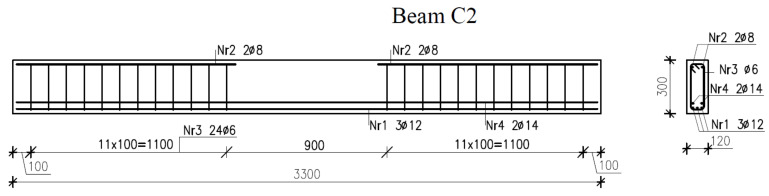
Reinforcement in beams C2.

**Figure 5 materials-14-03043-f005:**
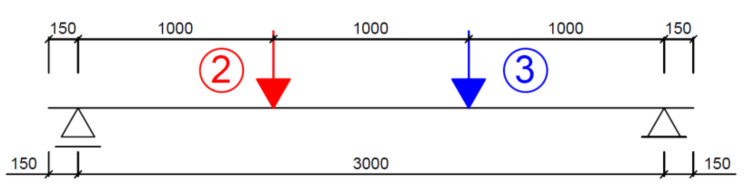
Static scheme of beams A2, D2 and C2.

**Figure 6 materials-14-03043-f006:**
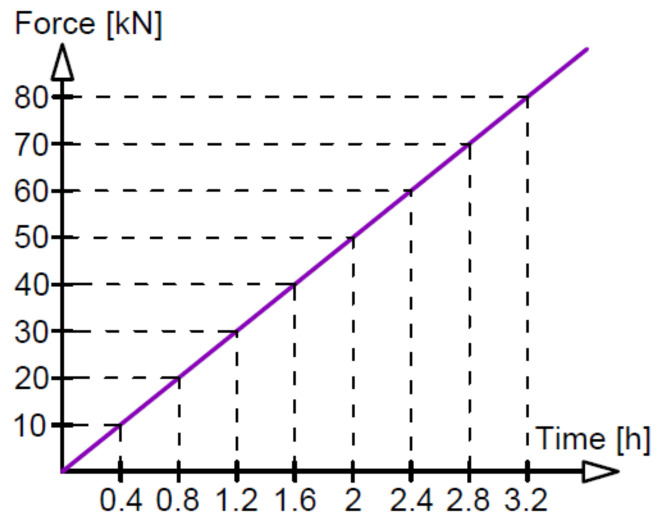
Monotonic loading.

**Figure 7 materials-14-03043-f007:**
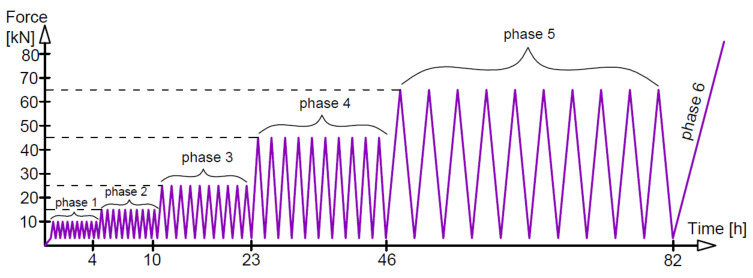
Loading and unloading.

**Figure 8 materials-14-03043-f008:**
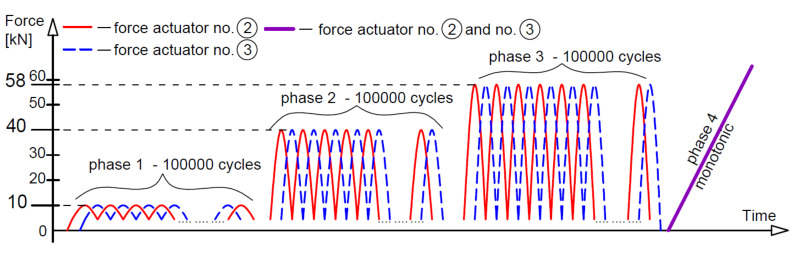
Low-cycle loading.

**Figure 9 materials-14-03043-f009:**
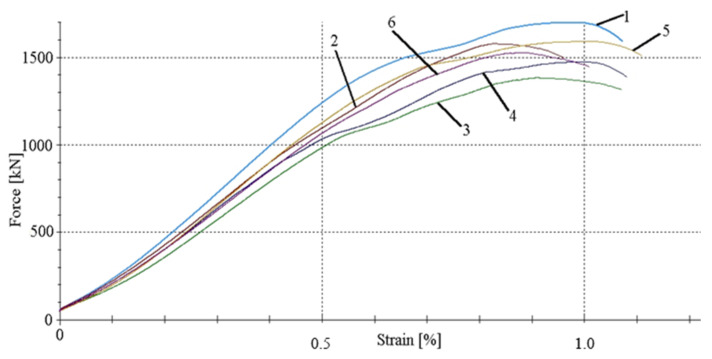
Force–strain relationships for concrete in beams A2M and A2O.

**Figure 10 materials-14-03043-f010:**
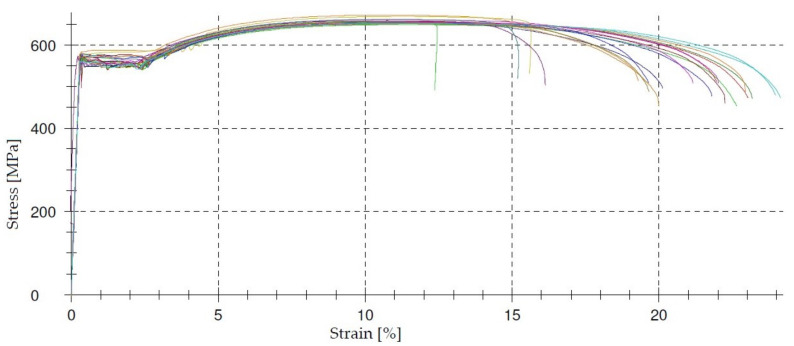
Stress–strain curve for 23 ϕ 12 bars.

**Figure 11 materials-14-03043-f011:**
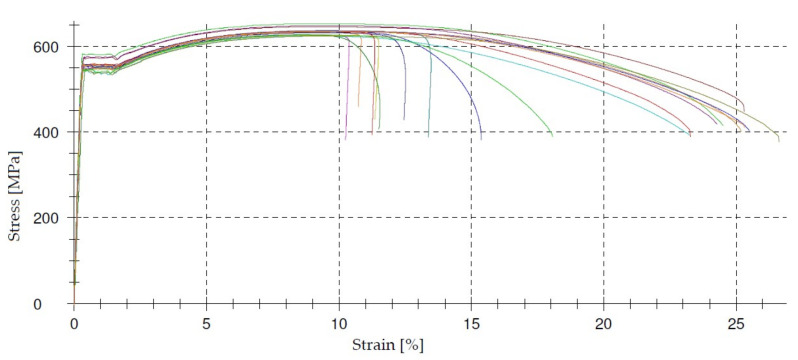
Stress–strain curve for 18 ϕ 14 bars.

**Figure 12 materials-14-03043-f012:**
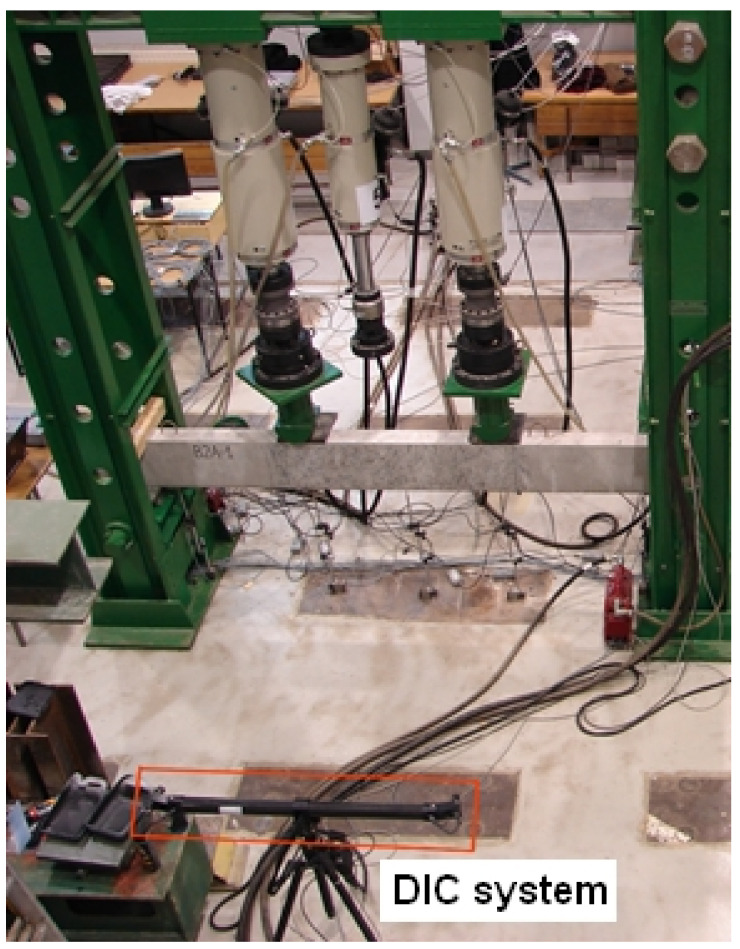
Photo of the test setup.

**Figure 13 materials-14-03043-f013:**
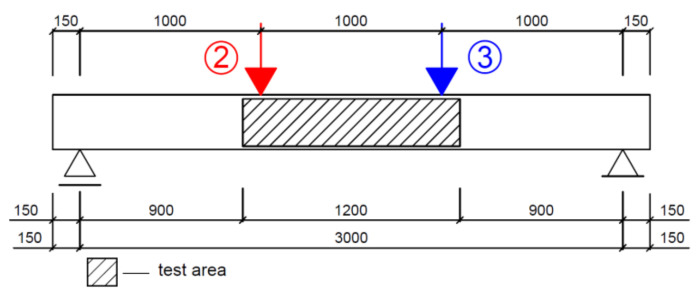
Loading scheme with marked test area.

**Figure 14 materials-14-03043-f014:**
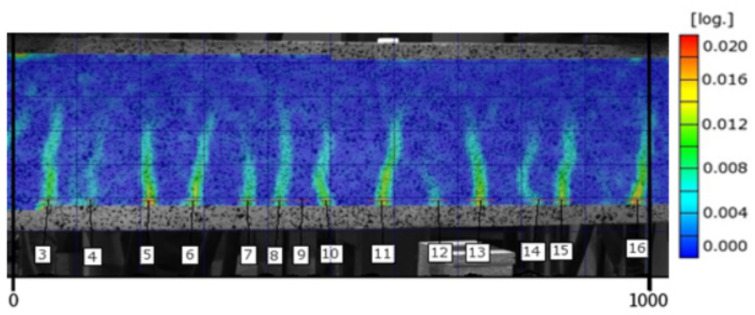
Strain map for beam C2M-1 at the load level of 0.75.

**Figure 15 materials-14-03043-f015:**
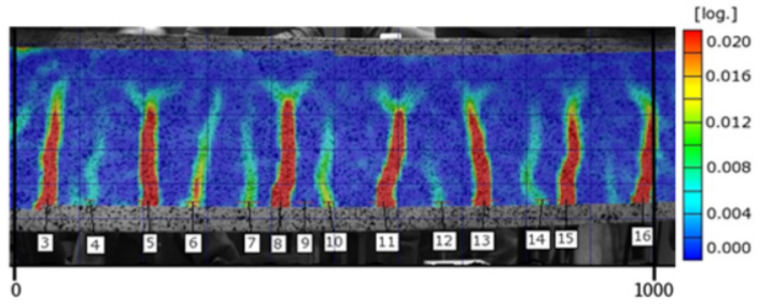
Strain map for beam C2M-1 at the load level of 0.95.

**Figure 16 materials-14-03043-f016:**
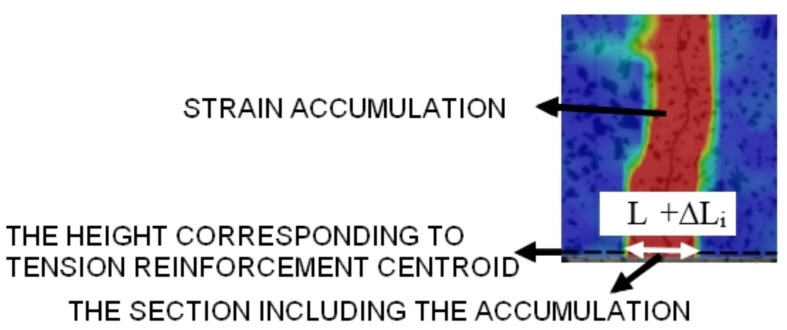
Crack width measurement scheme from DIC system.

**Figure 17 materials-14-03043-f017:**
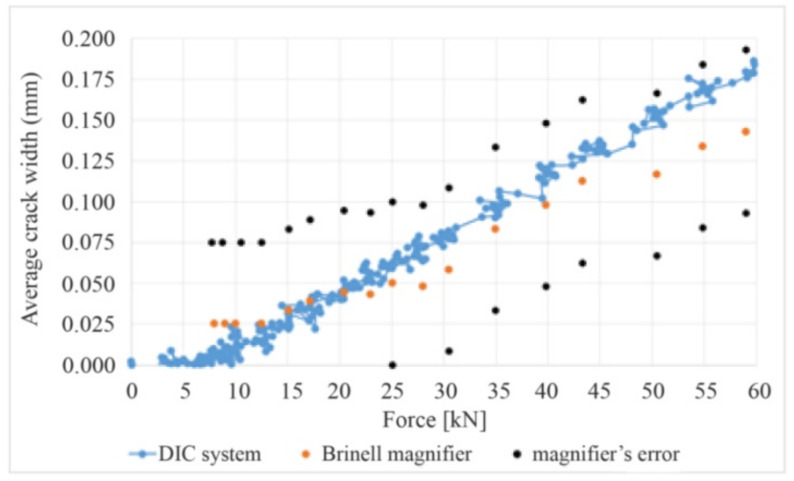
Average crack width from two measurement methods.

**Figure 18 materials-14-03043-f018:**
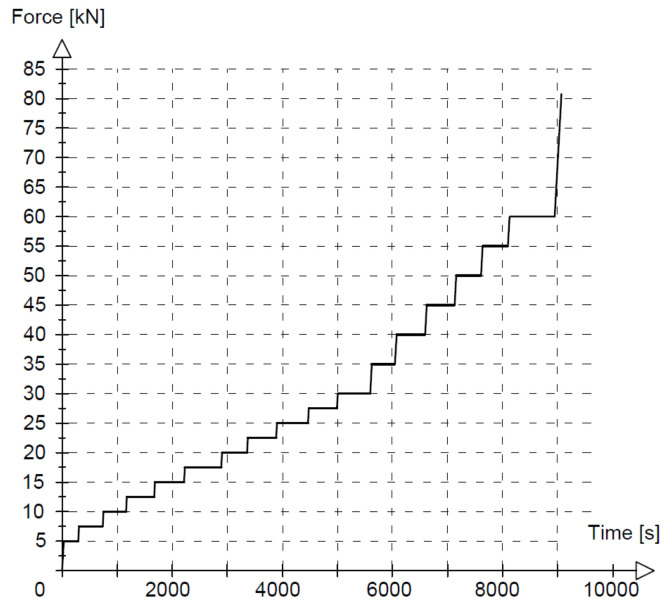
Loading with measurement intervals.

**Figure 19 materials-14-03043-f019:**
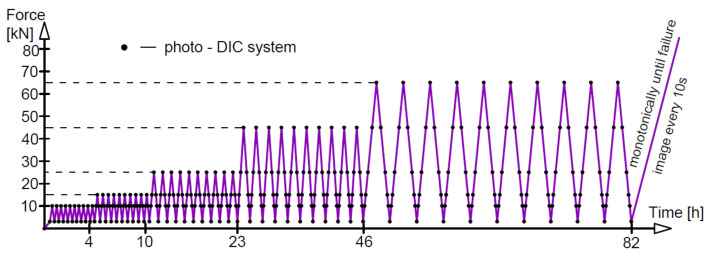
Digital imaging scheme for beams A2O.

**Figure 20 materials-14-03043-f020:**
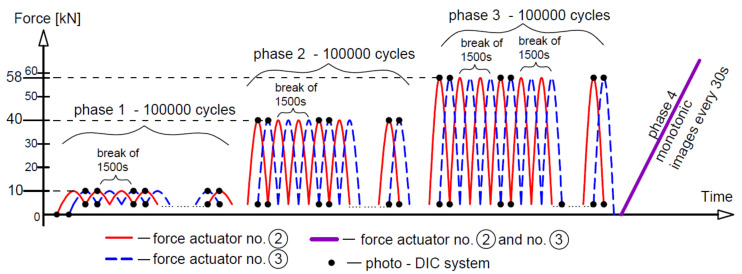
Digital imaging scheme for beams A2C.

**Figure 21 materials-14-03043-f021:**
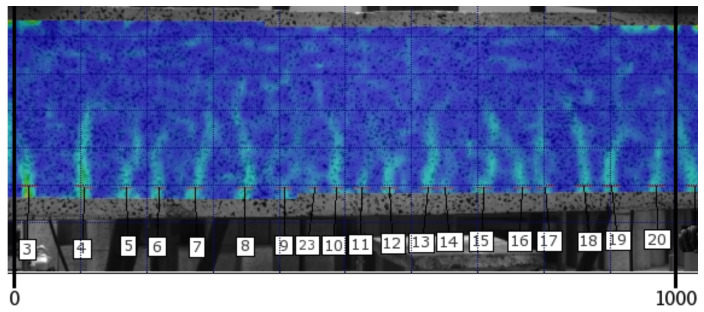
Strain map of the middle section of beam A2M-2 at the load level of 0.75.

**Figure 22 materials-14-03043-f022:**
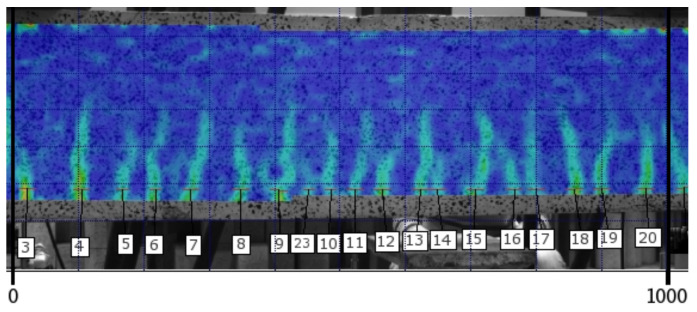
Strain map of the middle section of beam A2M-2 at the load level of 0.95.

**Figure 23 materials-14-03043-f023:**
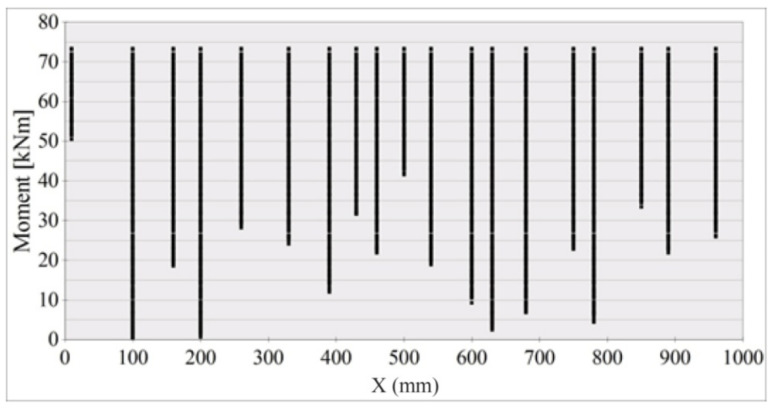
Crack formation and density under loading—beam A2M-2.

**Figure 24 materials-14-03043-f024:**
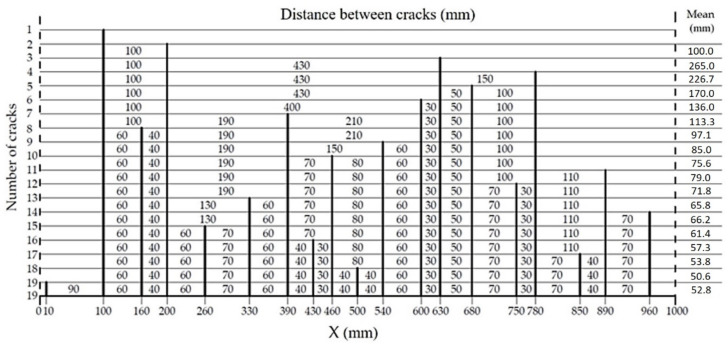
Summary of distances between cracks while loading beam A2M-2.

**Figure 25 materials-14-03043-f025:**
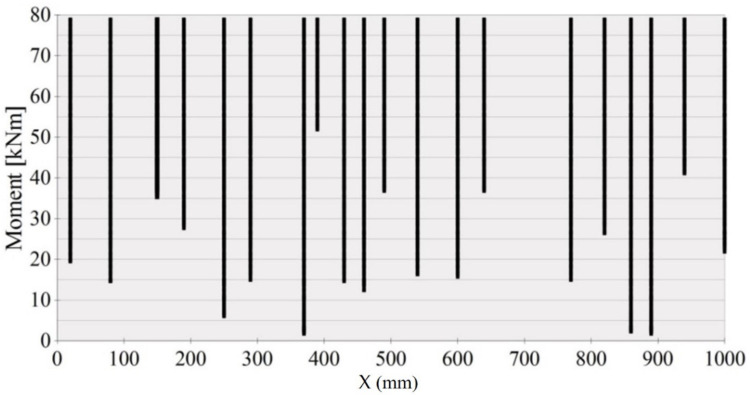
Crack formation and density under loading—beam A2M-1.

**Figure 26 materials-14-03043-f026:**
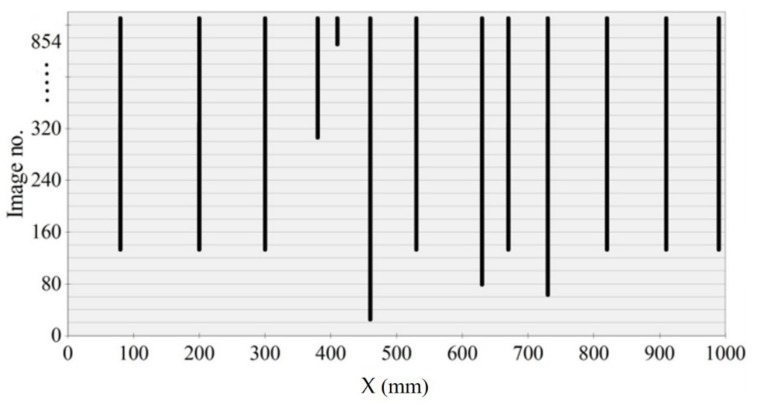
Crack formation and density under loading—beam A2C-1.

**Figure 27 materials-14-03043-f027:**
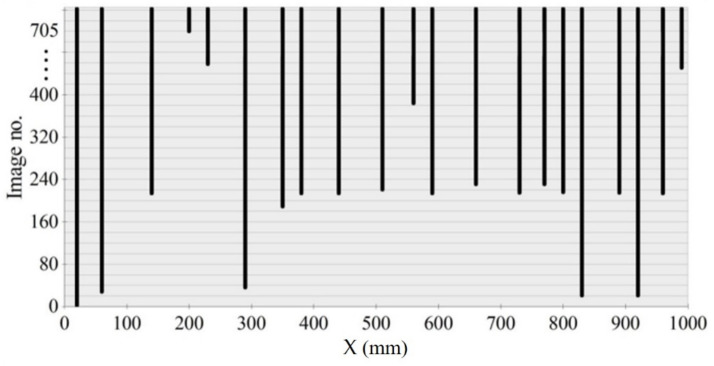
Crack formation and density under loading—beam A2C-2.

**Figure 28 materials-14-03043-f028:**
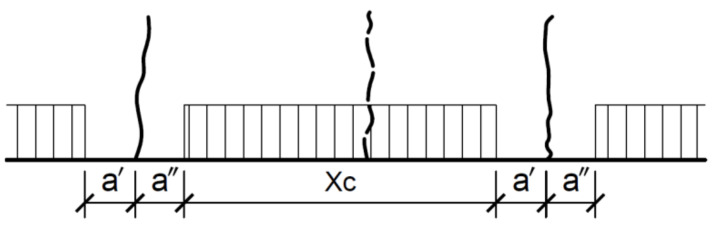
Model of crack formation—passive section **a** and active section **x_c_**.

**Figure 29 materials-14-03043-f029:**
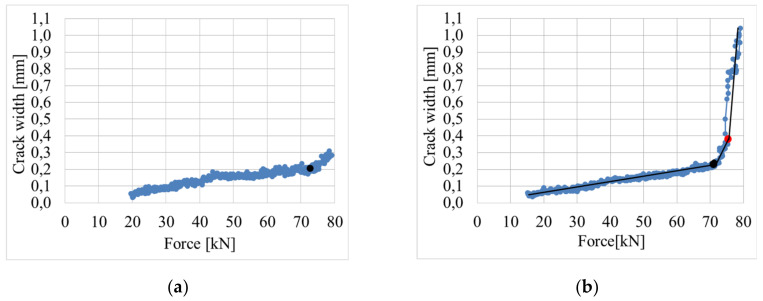
Increase in the width of cracks on beam D2M-2 during loading (**a**) crack no. 2; (**b**) crack no. 14.

**Figure 30 materials-14-03043-f030:**
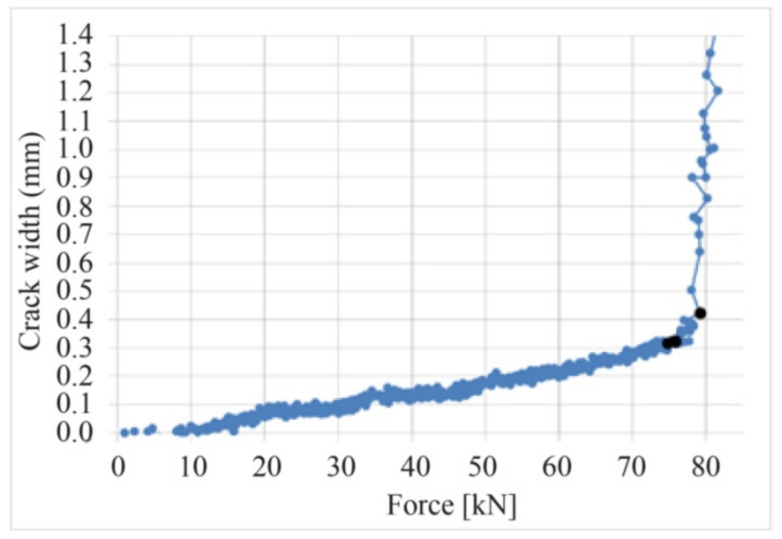
Increase in the width of crack no. 11 on beam C2M-2 during loading.

**Figure 31 materials-14-03043-f031:**
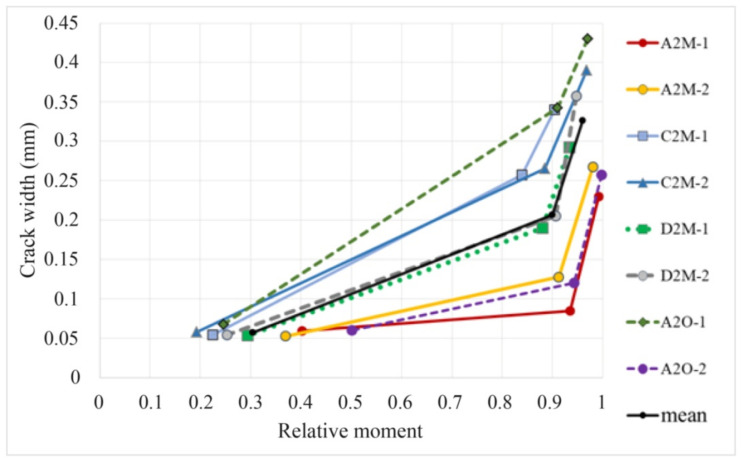
Increase in the mean crack width on individual beams under loading.

**Figure 32 materials-14-03043-f032:**
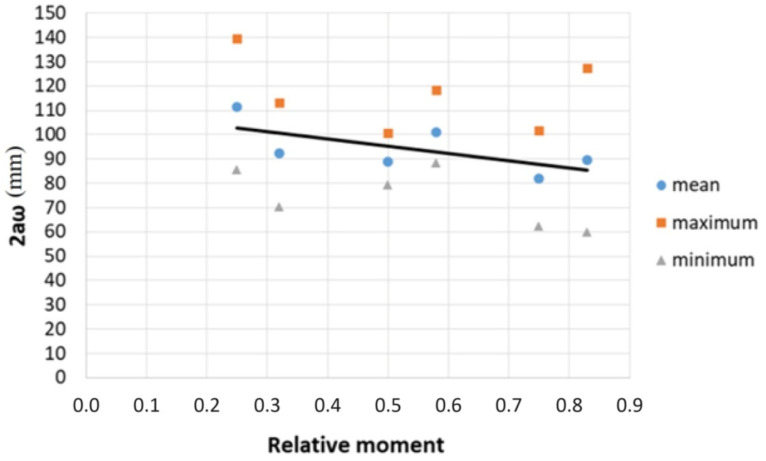
Section **2aω** by relative moment.

**Figure 33 materials-14-03043-f033:**
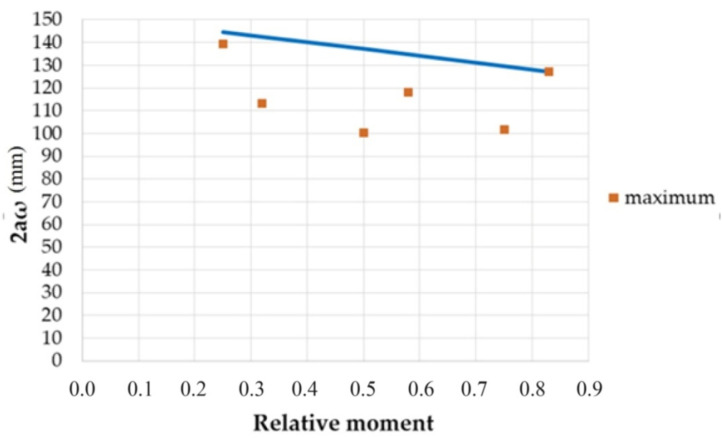
Maximal section **2aω** by relative moment.

**Figure 34 materials-14-03043-f034:**
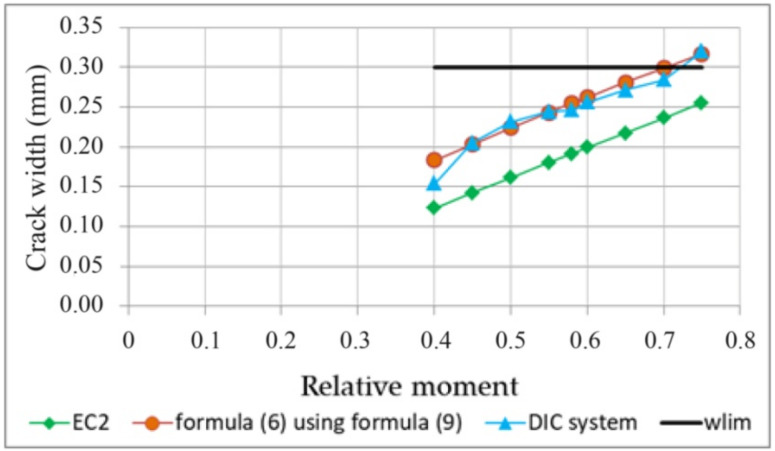
Crack widths at selected levels of critical stress–strain state.

**Table 1 materials-14-03043-t001:** Summary of beam data.

Reinforcement Design Symbol and Ratio	Reinforcement			Loading Program	Number of Beams
	Tension Reinforcement	Compression Reinforcement	Stirrups		
A2	3ϕ12—first row 2ϕ14—second row	2ϕ8	+	M—monotonic	2
A2		2ϕ8	+	O—with unloading	2
A2		2ϕ8	+	C—low-cycle	2
C2		None in the midspan section of the beam (ca.1m)	None in the midspan section of the beam (ca.1m)	M—monotonic	2
D2		2ϕ8	None in the midspan section of the beam (ca.1m)	M—monotonic	2

**Table 2 materials-14-03043-t002:** Loading program and rate for the members under test.

Reinforcement	Loading Regime	Loading Rate
A2	M—monotonic	0.4 kN/min
A2	O—loading and unloading	0.6 kN/min
A2	C—low-cycle	Phase 1—1 Hz/s Phase 2—0.5 Hz/s Phase 3—0.5 Hz/s Phase 4—0.4 or 0.6 kN/min
C2	M—monotonic	0.4 kN/min
D2	M—monotonic	0.4 kN/min

**Table 3 materials-14-03043-t003:** Summary of failure forces for concrete cubes.

Failure Force (kN)					
Beams		A2M, A2O	C2M	D2M	A2C
Specimen Number	1	1699.78	1699.16	1579.52	1598.35
	2	1581.12	1525.13	1571.93	1665.11
	3	1385.10	1509.84	1720.55	1640.01
	4	1473.84	1617.77	1590.06	1569.47
	5	1591.74	1607.50	1671.89	1722.58
	6	1527.79	1664.63	1571.92	1640.35

**Table 4 materials-14-03043-t004:** Compressive strength results for concrete cubes.

Compressive Strength (MPa)					
Beams		A2M, A2O	C2M	D2M	A2C
Specimen number	1	75.5	75.5	70.2	71.0
	2	70.3	67.8	69.9	74.0
	3	61.6	67.1	76.5	72.9
	4	65.5	71.9	70.7	69.8
	5	70.7	71.4	74.3	76.6
	6	67.9	74.0	69.9	72.9
Mean	$f_{\mathrm{cm}}^{\mathrm{cube}}$	68.6	71.3	71.9	72.9
Standard Deviation	s	4.8	3.3	2.8	2.4
Coefficient of Variation	ν	7.0%	4.7%	3.9%	3.2%

**Table 5 materials-14-03043-t005:** Concrete class.

Beams		A2M, A2O	C2M	D2M	A2C
$f_{\mathrm{cm}}^{\mathrm{CYL}}$	(MPa)	54.9	57	57.5	58.3
f_ck_	(MPa)	46.9	49	49.5	50.3
Class	-	C45/55	C50/60	C50/60	C50/60
E_cm_	(GPa)	36.7	37.1	37.2	37.3

**Table 6 materials-14-03043-t006:** Tensile test results for steel bars.

Statistics		Mean	Standard Deviation	Coefficient of Variation
		x	s	v
58 ϕ 12 mm				
E	GPa	197.17	3.59	1.82
R_p0.2_	MPa	567.2	8.84	1.56
R_eH_	MPa	570.89	7.49	1.31
R_m_	MPa	657.2	5.88	0.9
66 ϕ 14 mm				
E	GPa	201.41	5.13	2.55
R_p0.2_	MPa	558.47	11.98	2.15
R_eH_	MPa	561.83	12.67	2.26
R_m_	MPa	636.09	8.97	1.41
ϕ 12 and ϕ 14 (for all 124 bars)				
E	GPa	199.42	4.94	2.48
R_p0.2_	MPa	562.55	11.45	2.04
R_eH_	MPa	567.38	10.71	1.89
R_m_	MPa	645.96	13.05	2.02

**Table 7 materials-14-03043-t007:** Crack width results for beam A2M-2 (consecutive images).

Location X (mm)			100	160	200	260	330	390	460	500	540	600
Image No.	Actuator No.2 (kN)	Actuator No.3 (kN)	Crack No.									
			4	5	6	7	8	9	10	11	12	13
			Crack width (mm)									
212	43.08	42.98	0.147	0.085	0.065	0.102	0.075	0.027	0.055	0.051	0.083	0.092
213	43.14	43.05	0.162	0.035	0.074	0.107	0.087	0.038	0.069	0.030	0.102	0.037
214	42.26	42.00	0.143	0.066	0.070	0.099	0.076	0.039	0.046	0.028	0.104	0.078
215	41.67	41.54	0.129	0.039	0.071	0.088	0.096	0.028	0.082	0.028	0.105	0.082
216	43.23	43.04	0.129	0.068	0.063	0.073	0.075	0.041	0.089	0.030	0.095	0.052
217	43.54	43.35	0.140	0.082	0.068	0.072	0.092	0.040	0.083	0.026	0.103	0.057
218	43.36	43.24	0.114	0.089	0.071	0.081	0.092	0.041	0.053	0.041	0.105	0.080
219	43.44	43.31	0.122	0.079	0.070	0.088	0.088	0.044	0.037	0.039	0.095	0.108
220	44.40	44.18	0.130	0.079	0.085	0.090	0.083	0.034	0.097	0.062	0.102	0.089
221	44.95	44.85	0.120	0.086	0.073	0.098	0.093	0.037	0.054	0.052	0.111	0.060
222	44.81	44.75	0.096	0.107	0.074	0.080	0.069	0.062	0.076	0.051	0.119	0.077
223	43.59	43.43	0.107	0.061	0.064	0.082	0.071	0.043	0.096	0.040	0.108	0.060
224	45.58	45.37	0.096	0.076	0.064	0.101	0.087	0.051	0.100	0.027	0.111	0.072

**Table 8 materials-14-03043-t008:** Crack width results for beam A2M-2 at 0.75 and 0.95 of the actual damage force.

Image No.	Actuator No. 2 (kN)	Actuator No. 3 (kN)	Crack No.									
			4	5	6	7	8	9	10	11	12	13
			Crack width (mm)									
270	53.39	53.31	0.124	0.102	0.087	0.111	0.107	0.058	0.091	0.049	0.115	0.110
345	68.44	68.26	0.230	0.100	0.109	0.134	0.150	0.184	0.135	0.086	0.154	0.120

**Table 9 materials-14-03043-t009:** Location of the first crack and the value of the cracking moment for beams in bending.

Beam	A2M-1	A2M-2	D2M-1	D2M-2	C2M-1	C2M-2	A2O-1	A2O-2
X (mm)	370 and 890	100	680	890	870	960	20	710
Cracking moment (kNm)	1.87	0.67	8.62	14.49	11.87	4.16	3.23	9.66
Relative cracking moment	0.024	0.009	0.102	0.183	0.138	0.051	0.04	0.13

**Table 10 materials-14-03043-t010:** Maximum, minimum and mean value of 2aω for maximum crack width.

Relative Moment	2aω (mm)							
	A2M-1	A2M-2	D2M-1	D2M-2	A2O-1	Maximum	Minimum	Mean
0.25	132.77	139.38	87.40	85.67		139.38	85.67	111.30
0.32	70.17	74.76	96.34	106.59	113.07	113.07	70.17	92.19
0.5	83.42	79.41	100.40	91.60		100.40	79.41	88.71
0.58	96.47	88.45	102.28	98.57	118.26	118.26	88.45	100.80
0.75	62.51	71.75	91.72	101.68		101.68	62.51	81.91
0.83	59.93	63.65	97.81	99.08	127.04	127.04	59.93	89.50

**Table 11 materials-14-03043-t011:** Summary of the crack width calculations at selected load levels.

Crack Width (mm)								
Beam	Relative Moment							
	0.4	0.45	0.5	0.55	0.6	0.65	0.7	0.75
A2M-1	0.17	0.189	0.208	0.226	0.244	0.261	0.278	0.294
A2M-2	0.164	0.182	0.2	0.218	0.235	0.252	0.268	0.284
D2M-1	0.182	0.203	0.223	0.242	0.261	0.28	0.298	0.315
D2M-2	0.175	0.195	0.214	0.233	0.251	0.27	0.287	0.303
C2M-1	0.183	0.203	0.224	0.243	0.262	0.281	0.299	0.317
C2M-2	0.175	0.194	0.214	0.232	0.251	0.268	0.286	0.303
Mean	0.175	0.194	0.214	0.232	0.251	0.269	0.286	0.303

## Data Availability

Data available in a publicly accessible repository.

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
