# Peer review of "Analysis of Crack Width Development in Reinforced Concrete Beams"

_materials, 2021, doi:10.3390/ma14113043_

Round 1

Reviewer 1 Report

The manuscripts reports on an experimental study aimed at studying the onset and propagation of cracks in reinforced concrete beams in bending by using the digital immage correlation (DIC) technique.

The study has been correctly designed and conducted. The manuscript is interesting and worth of being published.

However, English language could benefit from a thorough revision, possibly by a native English speaker.

A few suggestions for minor revisions follows:
- line 12: "a structure" --> "concrete structures" (crack width may be less relevant for other construction materials);
- line 16: "low-cyclic loads" --> "low-cycle fatigue loads";
- lines 17-19: this sentence is quite long: please, try to reformulate by breaking it into shorter ones;
- line 22: "calculated to" --> "calculated according to";
- line 134: "mm" --> "mm^3";
- line 135: "concreting" --> "casting" (?);
- Table 4: for the sake of clarity, please define the meaning of each statistical quantity;
- line 257: "and the first quantity does not depend on the moment." --> this sentence sounds quite obvious (or not?) and probably could be deleted;
- line 298: "Deterministic determination of crack formation criteria leads" --> "Deterministic crack formation criteria lead";
- line 300: "the values of the cracking moment" --> "the critical value for cracking";
- lines 303-306: this sentence is quite long: please, try to reformulate by breaking it into shorter ones;
- lines 351-353: this sentence is quite long: please, try to reformulate by breaking it into shorter ones.

Author Response

All changes and additions are marked with red in the latest uploaded revised (according to reviewer comments) version text.
And the enclosed file is the brief response to the reviewers' comments in tabular form.

Reviewer 2 Report

The originality and the scientific value of the subject research are good.
The research area is Analysis of Crack Width Development in Reinforced Concrete Beams.

It is necessary to rewrite the abstract.
The abstract must engage the reader, specify the solved area, describe the experiments and briefly provide new knowledge.

The manuscript has the usual structure, but part of the discussion is missing!

References are incorrectly numbered and the References chapter must be modified according to the MDPI template.

Also in the case of reinforced concrete beams with shear reinforcement, it is very important to determine the detailed mechanical properties of concrete, where the influence of single-unit mechanical properties of concrete can be very well illustrated by numerical modeling or comparative tests of experiments.

It needs to improve figures: 2,3,4 - increase the font, improve the clarity of scheme

Clearly state the concrete recipe (cement, water, aggregate).

In laboratory tests, were only compressive strength tests performed?
Concrete is a quasi-brittle material. It is appropriate to use a comprehensive description of mechanical properties, where there is compressive strength, modulus of elasticity, tensile strength and fracture-mechanical properties.

Table 4. has bad formatting.

Variables are in italics. - Full manuscript, please

Figure 22. please enlarge.
Tensile tests were done only for reinforcement with a diameter of 8 mm?
What was the declared type B500? A B C

State the mechanical properties of the remaining (other diameters) reinforcement.

The label is on the wrong side. - 3. Test setup and instrumentation 

Table 9. - also state the value of the load in KN - not only kNm

Table 1 shows a total of 10 test beams.
Provide (Add)  load-displacement diagrams from tests from the testing machine !

Part of the discussion is missing.

Overall, it is necessary to process the manuscript with greater interest.

Extensive research is underway in the area of concrete structures and concrete when it is necessary to rework and expand the information in the introduction section. 
These are mainly the possibilities of determination of material properties of concrete, experimental testing, approaches to the choice of parameters, or taking into account the uncertainties in the calculation for the stochastic character of concrete.
Sucharda, O. et.al. Non-linear analysis of an RC beam without shear reinforcement with a sensitivity study of the material properties of concrete. Slovak J. Civil Eng. 2020, 28, 33–43.
Valikhani, A.et. al. Numerical Modelling of Concrete-to-UHPC Bond Strength. Materials 2020, 13, 1379

The discussion chapter must be presented separately and present the results in the context of current research. What is the same, what is different?

Overall, it is necessary to improve the presentation of the results and increase the informative value of the results.

The manuscript must be revised.

Author Response

(The authors gave the same response as above.)

Reviewer 3 Report

This paper analyzed the crack width development in reinforced concrete beams. Some modifications are mentioned here to enhance the quality of the manuscript:
1.  In page 1 line 2, it should be “…calculated according to 22 Eurocode 2.”
2.   Most of the references in this paper are dated and it is suggested to add some papers on crack or pore structure analysis of concrete in the last two years. For example, [1-2]

[1] Li L, Cao M, Xie C, et al. Effects of CaCO3 whisker, hybrid fiber content and size on uniaxial compressive behavior of cementitious composites[J]. Structural Concrete. 2019, 20(1): 506-518.

  1. Format error: many references have two numbers.
    4.  Fractal dimension is a good tool for crack analysis. I think the author can refer to this literature [3] to analyze the fractal dimension of cracks.

[2] Li L, Li Z, Cao M, et al. Nanoindentation and Porosity Fractal Dimension of Calcium Carbonate Whisker Reinforced Cement Paste After Elevated Temperatures (up to 900℃)[J]. Fractals. 2021, 29(2): 2140001.

[3] Zhang C, Han S, Hua Y. Flexural performance of reinforced self-consolidating concrete beams containing hybrid fibers[J]. Construction and Building Materials. 2018, 174: 11-23.

  1. What do the numbers 1-6 in Figure 9 represent? 
    6.  In introduction section, the application of DIC in the research of concrete crack is not sufficient.
  2. Formula numbers and figure numbers should not be used in the conclusion.
  3. The format of the conclusion section is incorrect.
  4. Except for the use of DIC, the study in this paper is very general.It is suggested that the author give a comparison with other studies in order to better reflect the innovation of this paper.

Author Response

(The authors gave the same response as above.)

Round 2

Reviewer 2 Report

The changes made the improvement of the manuscript.

The research area and results are from the context of the manuscript can better understand.

Please check the final version of the manuscript carefully according to the MPDI template.
For example:
reference [10], [11], [12] abbreviation str. - delete,
[14] character :  - delete
[15] add all authors
[16] character :  - delete
[24] abbreviation vol., pp. - delete

The results of the research and information value of the manuscript can be evaluated overall well.

The manuscript can be published in the journal.